# The Mbam drainage system and onchocerciasis transmission post ivermectin mass drug administration (MDA) campaign, Cameroon

Raphael Awah Abong[1,2], Glory Ngongeh Amambo[1,2], Ali Ahamat Hamid[1,2], Belinda Agbor Enow[1,2], Amuam Andrew Beng[1,2], Franck Noel Nietcho[1], Theobald Mue Nji[2,3], Abdel Jelil Njouendou[1,4], Manuel Ritter[5], Mathias Eyong Esum[1,2], Kebede Deribe[6,7], Jerome Fru Cho[1,2], Fanny Fri Fombad[1,2], Peter Ivo Enyong[1,2], Catherine Poole[8], Kenneth Pfarr[5,9], Achim Hoerauf[5,9], Clotilde Carlow[8], Samuel Wanji[1,2]*

1 Parasites and Vector Biology Research Unit (PAVBRU), Department of Microbiology and Parasitology, University of Buea, Buea, Cameroon, 2 Research Foundation in Tropical Diseases and Environment (REFOTDE), Buea, Cameroon, 3 Department of Sociology and Anthropology, University of Buea, Cameroon, 4 Department of Biomedical science, Faculty of Health Sciences, University of Buea, Buea, Cameroon, 5 Institute for Medical Microbiology, Immunology and Parasitology, University Hospital Bonn, Bonn, Germany, 6 Global Health and Infection Department, Brighton and Sussex Medical School, Brighton, United Kingdom, 7 School of Public Health, Addis Ababa University, Addis Ababa, Ethiopia, 8 New England Biolabs, Ipswich, Massachusetts, United States of America, 9 German Center for Infection Research (DZIF), partner site Bonn-Cologne, Bonn, Germany

* swanji@yahoo.fr

**Data Availability Statement:** All relevant data are within the manuscript and its Supporting information files.

## Abstract

### Background

The impact of large scale Mass Drug Adminstration (MDA) of ivermectin on active onchocerciasis transmission by *Simulium damnosum*, which transmits the parasite *O. volvulus* is of great importance for onchocerciasis control programmes. We investigated in the Mbam river system area, the impact of MDA of ivermectin on entomological indices and also verify if there are river system factors that could have favoured the transmission of onchocerciasis in this area and contribute to the persistence of disease. We compared three independent techniques to detect *Onchocerca* larvae in blackflies and also analyzed the river system within 9 months post-MDA of ivermectin.

### Method

*Simulium* flies were captured before and after 1, 3, 6 and 9months of ivermectin-MDA. The biting rate was determined and 41% of the flies dissected while the rest were grouped into pools of 100 flies for DNA extraction. The extracted DNA was then subjected to O-150 LAMP and real-time PCR for the detection of infection by *Onchocerca species* using pool screening. The river system was analysed and the water discharge compared between rainy and dry seasons.

**Funding:** The study was supported by DFG, https://www.who.int/tdr/capacity/en/, grant number: HO 2009/10-1; HO2009/14-1; HU2144/3-1, grant recipient: SW; by UNICEF/UNDP/World Bank/SHO Special Programme for Research and Training in Tropical Diseases (TDR), www.who.int/tdr/grants/en/, grant number: B40134. Grant recipient: SW. Lines 603-606. The funders had no role in study design, data collection and analysis, decision to publish, or preparation of the manuscript.

**Competing interests:** The authors have declared that no competing interests exist.

## Principal findings

We used human landing collection method (previously called human bait) to collect 22,274 adult female *Simulium* flies from Mbam River System. Of this number, 9,134 were dissected while 129 pools constituted for molecular screening. Overall biting and parous rates of 1113 flies/man/day and 24.7%, respectively, were observed. All diagnostic techniques detected similar rates of *O. volvulus* infection ($P = 0.9252$) and infectivity ($P = 0.4825$) at all monitoring time points. *Onchocerca ochengi* larvae were only detected in 2 of the 129 pools. Analysis of the river drainage revealed two hydroelectric dams constructed on the tributaries of the Mbam river were the key contributing factor to the high-water discharge during both rainy and dry seasons.

## Conclusion

Results from fly dissection (Microscopy), real-time PCR and LAMP revealed the same trends pre- and post-MDA. The infection rate with animal *Onchocerca sp* was exceptionally low. The dense river system generate important breeding sites that govern the abundance of *Simulium* during both dry and rainy seasons.

### Author summary

The presence of parasite strains that respond sub-optimally to an approved drug, favourable breeding sites for the vector and infected individual in an area, will surely provide conditions for continuous and persistent transmission of a disease despite a long-term control intervention. We investigated the impact of ivermectin on entomological indices within 9 months following a large-scale MDA in the Mbam river drainage. The river system factors that could have favoured abundance vector breeding and contribute to the persistence of disease transmission were also examined within the study period. We observed vector abundance and high entomological indices throughout the study period following ivermectin MDA. We also observed high water discharge along the main river of the drainage basin in both the rainy and dry seasons and this is due to the presence of two dams constructed upstream at Bamendjing and Mape to regularize the course of river Sanaga in view of generating hydroelectric power at Edea. Factors favouring continuous and persistent disease transmission are present in this drainage basin despite over 20 years of annual IVM-MDA. There is need for alternative control strategy in order to accelerate the fight against onchocerciasis in the area.

## Introduction

Onchocerciasis is a parasitic disease caused by *Onchocerca volvulus*, transmitted by the bite of an infective female black fly of the genus *Simulium* [1]. Worldwide, the disease is recognized as the second leading infectious cause of blindness after trachoma [2] and is a major public health concern in many tropical countries [3]. It is one of the neglected tropical diseases with an estimated 21 million people infected and about 99% of them are in Africa [4]. The Global Burden of Disease Study estimated in 2017 that, there were 20.9 million prevalent *O. volvulus* infections worldwide with 14.6 million of them suffering from skin disease and 1.15 million had vision loss [3]. In Cameroon, onchocerciasis is still endemic and is in urgent need of

intensification or alternative strategies to complement the current elimination programme [5]. Over the years, studies have depicted the situation in different bio-ecological zones in the country [6–11].

The control/elimination of onchocerciasis today relies on ivermectin (Mectizan) mass drug administration (IVM-MDA) to at risk populations in endemic countries. The impact of large scale IVM-MDA on *Simulium damnosum*, which transmits *O. volvulus* is of great importance for onchocerciasis control programmes. Mectizan has been donated free of charge by Merck and Co., Whitehouse Station, NJ since 1987. Ivermectin must be repeatedly given once a year for more than 15 years (with at least 60% geographical and chemotherapeutic coverage of MDA) to successfully achieve the elimination of onchocerciasis by the year 2030 as desired by the World Health Organization [12]. The use of IVM (Mectizan) biannually led to the elimination of parasite transmission in eleven foci in the Americas [13], while in Africa, annual community directed treatment with IVM (CDTI) for 10 to 17 years in the savannah areas reported prevalence rates and microfilarial loads as well as transmission indices below the thresholds required for elimination [14,15].

If IVM is administered in large scale, there will be a potential reduction of microfilaria (Mf) in the human population as it is known that a single standard dose (150 µg/kg body weight) of IVM can induce 98–99% reduction in skin microfilarial (Mf) load of *O. volvulus* within one month of treatment [16]. Thus, the available Mf in the human population for possible uptake during a blood meal by blackflies should also drop drastically following MDA. This will lead to a potential reduction in transmission potential at least within the first 3 months of MDA as reported in earlier studies [17–19]. An evaluation of the impact of IVM-MDA showed a significant decrease (53.5% reduction) in the infection rate of the vector population within two months post treatment [17]. Also, a study by Remme and colleagues [18] reported a reduction in transmission rate of between 65% -85% during the first three post-treatment months while Trpis and others [19] observed that, the number of infected flies with developing larvae (L1, L2, L3 stages) of *O. volvulus* in treated areas was reduced by 93.4–95%, while the monthly transmission potential (MTP) decreased from 22.9 to 5.8 (74.6% reduction). During such situations of low microfilaridemia and reduced transmission potential, microscopy may be less sensitive for detection of *O. volvulus* infection in skin snips and blackflies. Also, the larvae of animal *Onchocerca* species (*O. ochengi* or *O. ramachandani*) that are transmitted by the same vector could be mistaken for *O. volvulus* as they are difficult to morphologically differentiate during dissection.

Currently, transmission and parasite presence or absence are determined by capturing and dissection of black flies or using PCR to detect parasite DNA in pools of blackflies [20] or employing the LAMP technology (a newly developed gene amplification method [21]), which is capable of detecting parasite DNA even if there was just one microfilaria/larva in a pool of 100 flies within 60 minutes [20]. Accurate detection of *O. volvulus* in the vectors is essential for assessing the transmission profile, deciding when to stop MDA, and monitoring recrudescence [22]. If there is the possibility that other *Onchocerca* species are also present, molecular techniques that differentiate *O. volvulus* from *O. ochengi* or *O. ramachandani* should be employed.

Following the adoption of IVM-MDA as a control/elimination strategy in Cameroon, it has been demonstrated that the Community Directed Treatment with Ivermectin (CDTI) project of Central Region at the level of Bafia had limited impact on onchocerciasis in the human population despite more than 15 years of repeated annual treatment with IVM [8]. In some of the communities, about 60% of individuals screened for onchocerciasis were still harbouring Mf in their skin [8]. The presence of infection in humans even after numerous rounds of IVM treatment means that there is continuous transmission by the vectors. The entomological indicators for onchocerciasis transmission in this area before and following MDA are unknown. Thus,

this study was designed to investigate if there are factors that could have favoured the abundance, vector capacity and efficiency of *Simulium damnusum* for the transmission of onchocerciasis in this area and contribute to the persistence of disease despite a long-term effort for CDTI implementation. Following IVM-MDA, we collected blackflies within 9 months to analyse the changes of entomological indicators during the observation period. In order to know the best diagnostic tool for monitoring infection in vectors, we compared 3 independent diagnostic techniques namely: microscopy, real-time polymerase chain reaction (PCR) and loop-mediated isothermal amplification (LAMP) for the detection of *Onchocerca* larvae in blackflies within the nine months post IVM-MDA. We also wanted to understand the Mbam River drainage system and report on the dynamics of water flow at different seasons of the year.

## Methods

### Ethics statement

The protocol for this study received ethical approval from the Cameroon National Ethics Committee for Human Health Research (Ref: N˚2015/09/641/CE/CNERSH/SP). Administrative clearance was also obtained from the Cameroon Ministry of Public Health (Ref: N˚48/L/MINSANTE/SG/DLMEP/PNLO). At the level of the study community, details on the benefits and potential risk of the study were explained to fly collection volunteers and the community leader using information sheets. Fly collectors were recruited from the community based on their own willingness to participate in the study. They signed a written consent form and were aware that they could withdraw at any time if they so decided.

### Study area

Blackflies were collected in the village of Biatsotsa situated along the bank of River Mbam in the Bafia Health District (HD) in the Mbam drainage basin. The Bafia HD is in the Mbam and Inoubou Division of the Centre Region of Cameroon. The district is located about 120 kilometers (km) north from the Centre Regional capital, Yaoundé and it belongs to the Centre 1 CDTI project area and has had over 20 rounds of annual CDTI but is still meso-endemic (weighted prevalence varied from 24.4 to 57.0% Mf and from 3.6 to 37.4% for nodule presence across the surveyed communities) for onchocerciasis [8]. The altitude of this forest savanna transition zone varies from 1100 to 1300 meters above sea level and lies between coordinates 4˚45′00″ north and 11˚14′00″ east. The main activities of the inhabitants of this river drainage are agriculture (mainly cocoa and cassava production), fishing and sand mining, all of which expose them to repeated *Simulium* bites. In this transmission zone, MDA coverage (expressed as number treated out of the total population) from inception of CDTI in the year 2000 up to the time of this study have improved steadily over the years as reported earlier [23,24]. However, Kamga and colleagues observed a lower therapeutic coverage than that reported during the 2014 MDA campaign in the study site [23]. As of the year 2007, WHO [24] reported a 99.5% geographical coverage of all 15 CDTI projects in Cameroon including the Centre 1 CDTI project where the Mbam drainage basin is situated. The therapeutic coverage in the Centre 1 CDTI project increased from as low as 42% in 2000 to above 75% in 2009, ranged between 75–80% from 2010 to 2014 and remained above 80% during the 2015, 2016 and 2017 treatment periods [23].

### Study design

Blackflies *(Simulium)* were collected at the bank of river Mbam in the village of Biatsota before (D0) and at 1(D30), 3(D90), 6(D180) and 9(D270) months after MDA (Fig 1). The biting rate

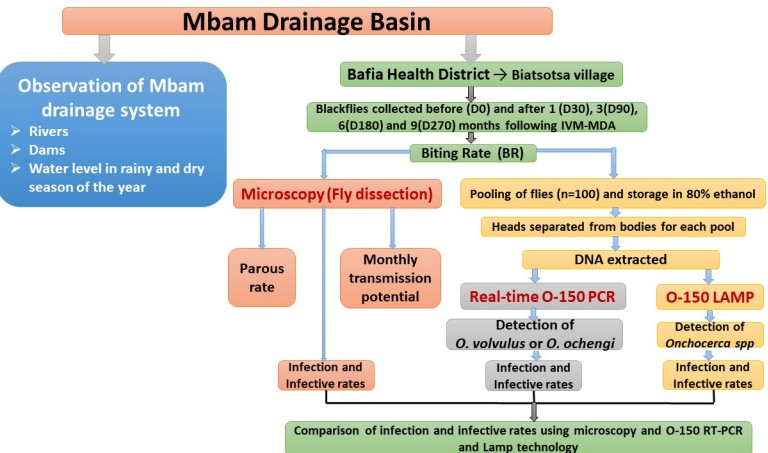

**Fig 1. Overview of the study design.**

was recorded and about 40% of the flies of each daily catch were dissected (microscopy) to compute the entomological indices and the rest grouped into pools of one hundred (that were later separated into sub pools of heads and bodies) and stored in 80% ethanol. DNA was extracted from the head and body pools for colorimetric O-150 LAMP as well as real-time PCR. The presence of *Onchocerca species* in blackflies was investigated and the sensitivity of the detection techniques compared at each time point.

## Collection of *Simulium* flies by human landing method

The fly collection team was composed of two trained individuals working alternately within 11 hours for each collection day, one working between 07:00 and 12:00 hours and the other between 12:00 and 18:00 hours [25]. Female *Simulium* flies coming for a blood meal were caught just as they landed using an aspirator as described in other studies [25,26]. Flies were collected by the same work force during the entire study.

## Dissection of female *Simulium* flies

Captured female blackflies were anaesthetized (knocked down) using chloroform, the species of fly was identified, counted and each placed in a drop of physiological saline on a clean glass slide and dissected under a dissecting microscope by an entomology technician. The dissection technique consists of, holding the fly with a needle in the thorax, piercing the posterior end of the abdomen with a dissecting needle and then pulling out the internal organs to determine the quantity of fat bodies, the state of the malpighian tubules and the ovaries. We could distinguish parous (very reduced or clear malpighian tubules and less or no fat bodies) from nulliparous (large or opec malpighian tubules and more fat bodies) flies as already described [27,28]. The abdomen, thorax and head of parous flies were further dissected and examined for the presence of *Onchocerca*. Any larvae (L1, L2 and L3) found were counted and recorded on a dissection sheet (S1 File). Undissected flies were counted into pools of 100 and stored in 80% ethanol for pool screening.

## Separation of *Simulium* flies into sub pools of heads and bodies

Using a dissecting microscope, fly heads were carefully separated from the body (abdomen + thorax) and stored separately as head pools or body pools. The separation technique

consisted of placing the fly in a Petri dish, holding it with a needle (0.5mL disposable insulin syringe) in the thorax, cutting off the head with the sharp edge of the needle and then placing them into separate tubes and stored at -80°C.

### Genomic DNA extraction from pools of *Simulium* flies

Each pool of the fly heads or bodies was weighed (average weight for head and body pools was 17 mg and 100 mg, respectively) and placed in a 1.5 mL Eppendorf tube. Total DNA was extracted from the head or body pools using the Zymo Research Genomic DNA Tissue Mini-Prep Kit (Epigenetics Compaty, USA), following manufacturers' protocol with slight modification in the case of body pools where the volumes of digestion buffer and proteinase K were doubled because the tissue mass was more than 25 mg. The eluted DNA (200 μL) from each pool was stored at -20°C.

### Detection of *O. volvulus* in pool-DNA of blackfly using colorimetric O-150 LAMP

The colorimetric O-150 LAMP assay was carried out essentially as previously described [20,29]. Amplification resulted in a colour change from pink to yellow in positive samples, while negative samples remained pink with no ambiguity in colour determination when read against a white background (white A4 paper) and results recorded (S2 File). Because of the high sensitivity of LAMP, DNA contamination and carry over of amplified products was prevented by using filter tips at all times, cleaning all work surfaces with 10% bleach solution before and after each session of work, performing each step of the analyses in separate work areas and minimizing manipulation of the reaction tubes. All tubes were tightly closed and were never opened to avoid contaminating the work area. The assay was optimized using DNA from *O. volvulus* microfilariae as a positive control and nuclease free water as the negative control. The negative controls never amplified while the positive controls always amplified. DNA samples from *M. perstans* and *L. loa* were also used to confirm the specificity of the test.

### Detection of *O. volvulus / O. ochengi* in pool-DNA of blackfly using real-time PCR

We carried out species-specific real-time PCR (OvOo Triplex real-time PCR) to detect the presence of *O. volvulus* and/or *O. ochengi* DNA in the *Simulium damnosum* vectors.

Prior to the assay, DNA samples from the black fly pools were tested for PCR inhibitory factors using the mouse IFN-γ real-time PCR as previously described [30], but no inhibition was detected (S1 Fig). *O. volvulus* DNA was detected in *Simulium* flies using a triplex PCR that differentiates *O. volvulus* from *O. ochengi* nematodes based on differences in the ND5 genes of these organisms (GenBank: AY462885.1 and FM206483.1, respectively) and included primers and hybridization probes for a consensus *Onchocerca spp.* 16S genes (all sequences available in GenBank in 2014). The PCR was conducted using 2 μL DNA in 10 μL volume in a Rotor Gene 6000 (Qiagen, Germany) with the following reaction mixture: 3.75 μL of nuclease free water, 1 μL of 10X PCR buffer, 1.2 μL of 25mM. MgCl$_2$; 0.05 μL of 10mM dNTPs mix, 0.3 μL of 10 μM OvOo ND5 Fw (5'-GCTATTGGTAGGGGTTTGCAT-3'), 0.3 μL of 10 μM OvOo ND5 Rev (5'-CCACGATAATCCTGTTGACCA-3'), 0.5 μM of 10 uM 16S Fw (5'-AATTACTCCG-GAGTTAACAGG-3'), 0.5 μL of 10 μM 16S Rev (5'-TCTGTCTCACGACGAACTAAAC-3'), 0.1 μL of 5 μM Ov probe (Fam-TAAGAGGTTATTGTTTATGCAGATGG-3'), 0.1 ML of 5 μM Oo probe (Hex-TAAGAGATTGTTGTTTATGCAGATAGG-3'), 0.15 μL of 5 μM 16S

probe (Cy5-GCTGCGCTACATCGATGTTGTA-3'), 0.05 μL of 5 Units HotStar Taq. Cycling conditions were as previously described [30]. Plasmids ($E^6$ copies/μL) containing the respective sequences were used as PCR positive controls in every run. Signals were analysed using Rotor-Gene Software version 2.3.1 (Build 49) with thresholds set to 0.01 for *O. volvulus*, 0.03409 for *O. ochengi*, 0.02026 for 16S (*Onchocerca species*) and an outlier removal of 10%. A positive signal (maximum $C_t$ value of 35) in the *O. volvulus* channel was considered positive for infection/exposure to *O. volvulus* while a positive signal in *O. ochengi channel* was considered positive (maximum $C_t$ value of 35) for *O. ochengi* infection and results recorded (S2 File). Every positive pool of bodies was interpreted as being infected with developing *O. volvulus* larvae or microfilariae, whereas a positive pool of heads was interpreted as containing infective L3H parasites. Pool screen algorithm [31] was used to estimate the infection and infective rates at 95% confidence interval.

### Generating the Mbam river drainage map

Data used was downloaded the Atlas Forestier Cameroun and the Qgis (Qgis version 2.18.11) software used to generate the Mbam river drainage. Dam and locality coordinates were collected using a global positioning system (GPS).

### Data management and analysis

The data collected were keyed into a template developed in Microsoft Excel 2013 and later exported to SPSS version 20 (IBM SPSS Statistics 22; Armonk, NY) for statistical analysis. Biting rate and other entomological indices from microscopy (parous, infection and infective rates, monthly transmission potential) were computed using electronic calculator as previously described [25]. The infection and I nfective rates from pool screening were computed using the algorithm described by Katholi and colleagues [31].

$$P = \sqrt[n]{k/m}$$

Where **m** = number of pools, **n** = size of pool, **k** = number of negative pools and *P* = prevalence of infection.

Kruskal-Wallis test was used to compare the average variations in infection and infective rates over the different time points for the different diagnostic techniques investigated.

## Results

### Dynamics in onchocerciasis transmission in *Simulium damnosum* flies caught within 9 months following IVM-MDA

A total of 22,274 black flies (*S. damnosum*) were collected and 9134 dissected to compute the entomological indices. The rest remaining 13,140 flies were stored in 80% ethanol and 12,900 of them were divided into 129 (D0 = 15, D30 = 17, D90 = 30, D180 = 14, and D270 = 53) pools for DNA extraction and detection of *Onchocerca* species.

### Entomological indices determined by microscopy (dissection)

Based on standard methods previously described [25], the entomological parameters by microscopy (dissection), at different time points (Day 0, Day 30, Day 90, Day 180 and Day 270) were determined (Table 1).

Before IVM-MDA commenced (Day 0), 4,806 flies were collected in four days resulting in a daily biting rate of 1,201 flies/man/day. A total of 2,535 flies were dissected and the parous rate was 18.9% (478/2,535), the infection rate was 25.1 flies per 1,000 parous flies, the infective rate

**Table 1. Entomological indices at different time points of collection before and after MDA.**

| Entomological survey | Day 0 | Day 30 | Day 90 | Day 180 | Day 270 | Total |
|---|---|---|---|---|---|---|
| Month and No. of captured flies | June 4806 | July 2916 | September 4559 | January 2297 | April 7696 | 22,274 |
| Number of fly collection days | 4 | 4 | 4 | 3 | 5 | 20 |
| Biting rate (flies/man/day) | 1201 | 729 | 1140 | 765 | 1539 | 1113 |
| No. of dissected flies | 2535 | 1380 | 1792 | 1271 | 2156 | 9134 |
| No. of parous females | 478 | 389 | 395 | 523 | 472 | 2257 |
| Parous rate (%) | 18.9 | 29.7 | 22.0 | 41.4 | 21.9 | 24.7 |
| No. of infected flies (L1, L2, L3) | 12 | 26 | 18 | 10 | 27 | 93 |
| Infection rate (%) | 2.5 | 6.68 | 4.56 | 1.9 | 5.7 | 4.12 |
| No. of infected flies per 1,000 parous flies | 25.1 | 66.8 | 45.6 | 19.1 | 57.2 | 41.2 |
| No. of females with L3 in the head | 7 | 6 | 3 | 3 | 6 | 25 |
| No. L3 in the head of infective flies | 24 | 14 | 5 | 11 | 22 | 76 |
| Infective rate (%) | 1.46 | 1.54 | 0.759 | 0.57 | 1.27 | 1.1 |
| Number of infective flies per 1,000 parous flies | 14.6 | 15.4 | 7.6 | 5.7 | 12.7 | 11.0 |
| Monthly transmission potential (L3H#/man/month) | 340 | 229 | 95 | 205 | 471 | |

# infective L3 larvae found in fly heads

was 14.6 infective flies per 1,000 parous flies and a Monthly Transmission Potential (MTP) of 340 infective L3 in the head transmitted into one person in that month (L3H/person/month) was calculated (rainy season).

One month after IVM-MDA, 2,916 flies were collected with a daily biting rate of 726 flies/man/day. The parous rate was 29.7% (389/1,380), the infection rate was 66.8 infections per 1,000 parous flies, the infective rate was 15.4 infective flies per 1,000 parous flies with an MTP of 229 L3H/person/month at the peak of rainy season.

Three months after IVM-MDA (September 2016), 4,559 flies were collected with a daily biting rate of 1,140 flies/man/day. The parous, infection and infective rates were 22.0% (395/1,792), 45.6 per 1,000 parous flies and 7.6 per 1,000 parous flies, respectively reducing the MTP to 95 L3H/man/month (end of rainy season).

Six months after IVM-MDA (January 2017), 2,297 flies were collected with a daily biting rate of 765 flies/man/day. The parous rate was 41.4% (523/1271), the infection rate was 19.1 per 1,000 parous flies, the infective rate was 5.7 per 1,000 parous flies and MTP was 205 L3H/man/month (dry season).

Nine months after IVM-MDA (April 2017) 7,696 flies were collected with a daily biting rate of 1,539 flies/man/day. The parous, infection and infective rates were 21.9% (472/2,156), 57.2 per 1,000 parous flies and 12.7 per 1,000 parous flies, respectively, with an MTP of 471 L3H/man/month (start of rainy season).

Overall, biting rate of 1,113 flies/man/day, a general parity of 24.7% (2257/9,134), infection rate of 41.2 per 1000 parous flies and an infective rate of 1.1 per 1,000 parous flies were observed.

## Dynamics of entomological parameters by microscopy

The dynamics in the biting rate, infection rate, infective rate and MTP of *Simulium* vectors at different time points before and after IVM-MDA determined by microscopy are illustrated below.

**a.) Biting rate dynamics.** The biting rate of *Simulium* vector for *O. volvulus* in the Mbam river basin fluctuates between 729 and 1,539 flies/man/day (Fig 2). The biting rates during the

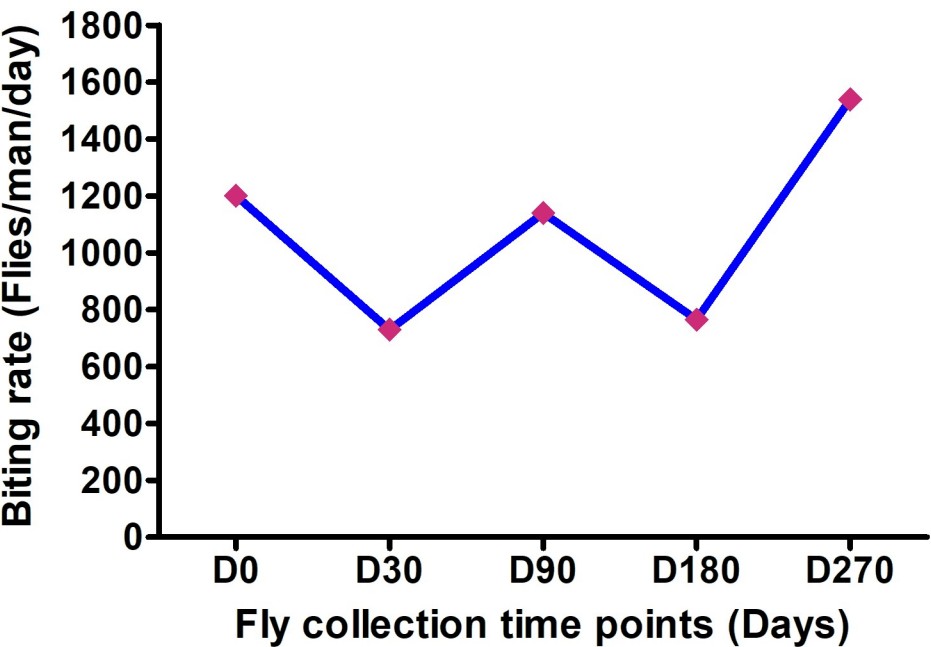

**Fig 2.** *Simulium* **biting rate at different time points within 9 months of MDA.**

rainy season (June or D0, July or D30 and September or D90) are not very different from those during the dry season (January or D180 and early April). On average, the biting rate was 1,113 flies/man/day.

**b.) Dynamics in the infection and infective rates.**    The infection rate was 25.1 per 1,000 parous flies before IVM-MDA and 66.8 per 1,000 parous flies at 30 days post control intervention (Fig 3). It later decreased gradually at 90 and 180 days post treatment but rose back to 57.2 per 1,000 parous flies at 270 days post treatment. The infection rate was above 19 per 1,000 parous flies at all collection time points pre- and post-treatment and in the rainy and dry seasons. The average infection rate in this area was 41.2 per 1,000 parous flies (Table 1).

Infective rate was also analysed over all observed time points (Fig 3). Before IVM-MDA, the infective rate was 14.6 per 1,000 parous flies and increased to 15.4 per 1,000 parous flies at 30 days post MDA but later decreased at 90 and 180 days after treatment. By day 270 post IVM-MDA, the infective rate had increased to 12.7 per 1,000 parous flies. On average the infectivity rate in this study was 11 per thousand parous flies (Table 1).

**c.) Dynamics in monthly transmission potential.**    The monthly transmission potential of *O. volvulus* infective larvae from flies to humans dropped from 340 L3H/man/month before IVM-MDA to 229 L3H/man/month after 30 days of treatment. The MTP further decreased to 95 L3H/man/month at D90 but drastically increased to 205 and 471 L3H/man/month at six and nine months post treatment, respectively (Fig 4). Within the first three months post MDA, MTP dropped from 340 to 95 (72% reduction).

## Detection of *O. volvulus* infection using colorimetric O-150 LAMP and real-time PCR

Given that fly heads were separated from bodies, we had a total of 258 sub pools (129 head pools and 129 body pools). With the pool screening technique, we determined two entomological parameters (Infection rates and Infective rates). To obtain our infection/infective rates per

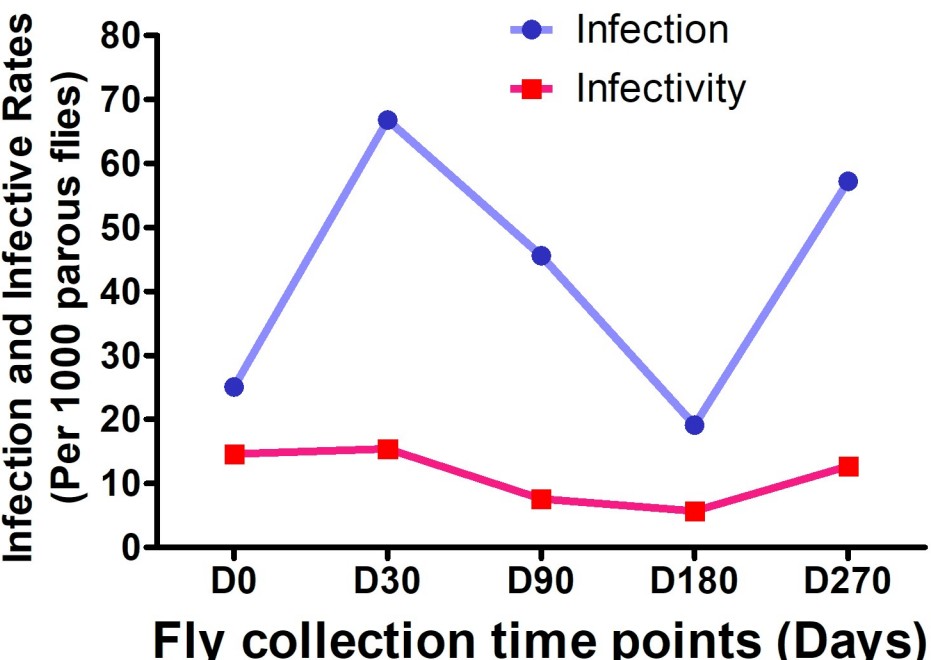

**Fig 3. Infection and infective rates dynamics within 9 months of MDA.**

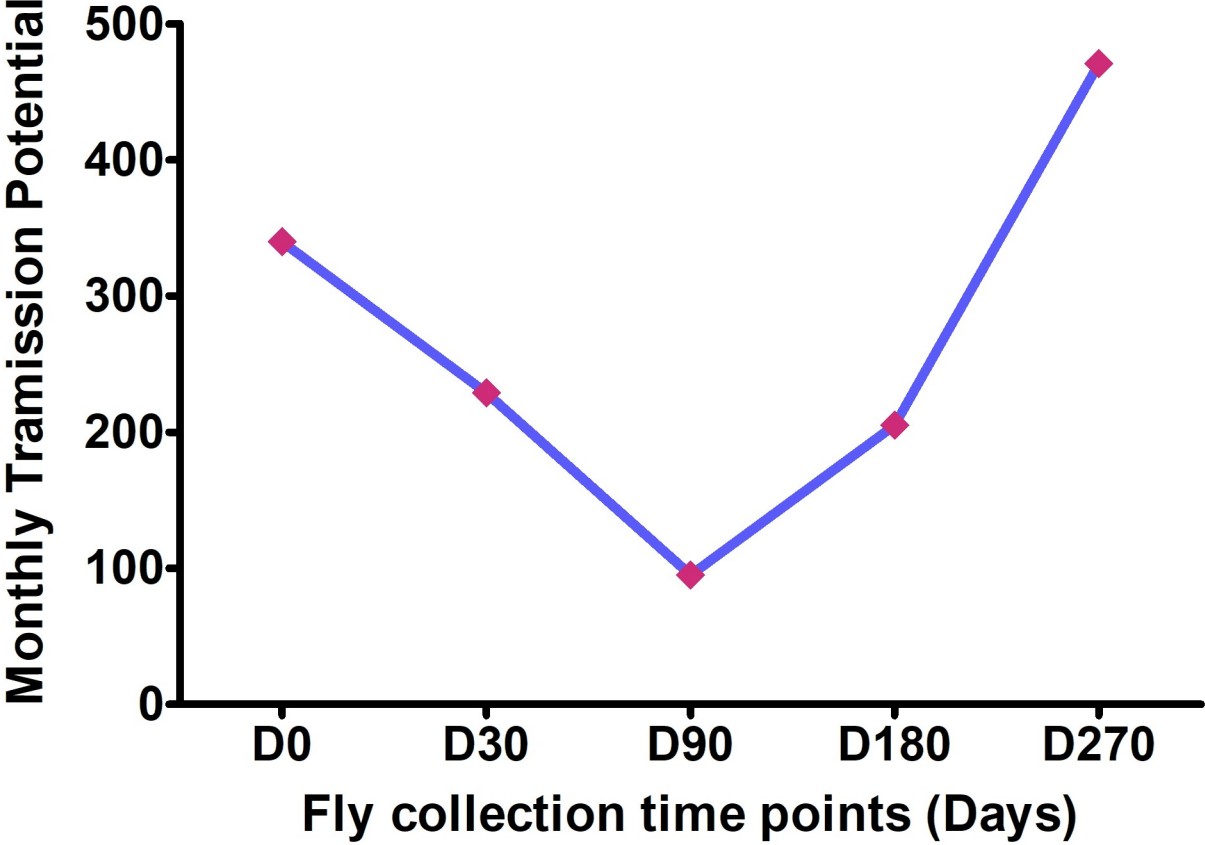

**Fig 4. Dynamics in the MTP within 9 months of IVM-MDA.**

1,000 parous flies for the pool screening, we multiplied calculated rates per thousand flies by 2 as recommended by the World Health Organization [32], assuming a parity of 50%.

## Colorimetric O-150 LAMP

LAMP reactions containing genetic material of *O. volvulus* that changed from pink to yellow were considered positive while negative reactions remained pink (S2 Fig). For this assay, 47 (36.4%) out of 129 head pools were positive while 114 (88.4%) of the 129 pools were positive for either body or head infection or both (Table 2). From the mathematical algorithm [31], colorimetric O-150 LAMP gave the infective and infection rates of 9.0 (n = 129 pools of 100 flies with 82 negative pools) and 42.6 (n = 129 pools of 100 flies with 15 negative pools) per 1,000 parous flies, respectively.

## Triplex real-time PCR

The triplex real-time PCR detected *O. volvulus* parasite DNA in 43 (33.3%) head pools while 118 (91.5%) of the 129 pools were positive for either body or head infection or both (Table 3). From the pool screening algorithm, the infective rate was 8.1 (n = 129 pools of 100 flies with 86 negative pools) per 1,000 parous flies while 48.6 (n = 129 pools of 100 flies with 11 negative pools) per 1,000 parous flies was determined as the infection rate.

## Comparing the sensitivity of microscopy to that of real-time PCR and O-150 LAMP in detecting *O. volvulus* infection within 9 months following IVM-MDA

As shown in Table 4, there was no statistically significant difference in the average variations in infection rates ($P = 0. 0.9252$) as well as the infective rates ($P = 0.4825$) among the different diagnostic tests at each time point, in detecting *O. volvulus* infection in black fly populations (Table 4). The pattern in infection rates were similar for the different diagnostic methods following 9 months of monitoring after MDA (Fig 5).

**Table 2. Infection and infective rates by pool screening using colorimetric O-150 LAMP technology.**

| Collection time point | No. of pools | Negative pools (Heads+Bodies) | Infection rate (%) | Negative pools (Heads only) | Infective rate (%) |
|---|---|---|---|---|---|
| D0 | 15 | 3 | 3.18 | 10 | 0.80 |
| D30 | 17 | 1 | 5.58 | 10 | 1.06 |
| D90 | 30 | 6 | 3.14 | 22 | 0.62 |
| D180 | 14 | 1 | 5.20 | 10 | 0.68 |
| D270 | 53 | 4 | 5.10 | 30 | 1.14 |
| **Total** | **129** | **15** | **4.26** | **82** | **0.90** |

**Table 3. Infection and infective rates by pool screening using triplex real-time PCR.**

| Collection time point | No. of pools | Negative pools (Heads+Bodies) | Infection rate (%) | Negative pools (Heads only) | Infective rate (%) |
|---|---|---|---|---|---|
| D0 | 15 | 4 | 2.62 | 11 | 0.62 |
| D30 | 17 | 1 | 5.58 | 13 | 0.54 |
| D90 | 30 | 3 | 4.56 | 19 | 0.92 |
| D180 | 14 | 2 | 3.86 | 10 | 0.68 |
| D270 | 53 | 1 | 7.78 | 33 | 0.94 |
| **Total** | **129** | **11** | **4.86** | **86** | **0.81** |

**Table 4. Comparison of infection and infective rates using microscopy, colorimetric LAMP and triplex real-time PCR.**

| Technique | Parameter | D0 | D30 | D90 | D180 | D270 | Total |
|---|---|---|---|---|---|---|---|
| **Fly dissection (Microscopy)** | Infection rate per 1,000 parous flies | 25.1 | 66.8 | 45.6 | 19.1 | 57.2 | **41.2**[*] |
| | Infective rate per 1,000 parous flies | 14.6 | 15.4 | 0.8 | 5.7 | 12.7 | **11.0**[#] |
| **Colorimetric O-150 LAMP for PoolScreening** | Infection rate per 1,000 parous flies | 31.8 | 55.8 | 31.4 | 52.0 | 51.0 | **42.6**[*] |
| | Infective rate per 1,000 parous flies | 8.0 | 10.6 | 6.2 | 6.8 | 11.4 | **9.0**[#] |
| **Triplex real-time PCR Assay for PoolScreening** | Infection rate per 1,000 parous flies | 26.2 | 55.8 | 45.6 | 38.6 | 77.8 | **48.6**[*] |
| | Infective rate per 1,000 parous flies | 6.2 | 5.4 | 9.2 | 6.8 | 9.4 | **8.1**[#] |

[*] **Infection rate**: Kruskal—Wallis statistics = 0.1556, **P** = 0.9252

[#] **Infective rate**: Kruskal—Wallis statistics = 1.458, **P** = 0.4825

### Detection of *O. ochengi* infection using real-time PCR

Using the triplex real-time PCR for the detection of *O. ochengi* parasites (larvae) in the *Simulium damnosum*, none of the 129 head pools were positive (0%) while two of the 129 body pools (1.6%) had positive signals. This gave us a zero infective rate and an infection rate of 1.4 (n = 129 pools of 100 flies with 127 negative pools) per 1,000 parous flies. The two pools that were positive for *O. ochengi* were collected at 9 months post MDA in April 2017.

### The Mbam drainage system

The Mbam drainage basin is irrigated by many fast-flowing rivers including the Mbam and the Noun rivers (Fig 6). The source of these rivers is in the highlands far from the Bafia Health

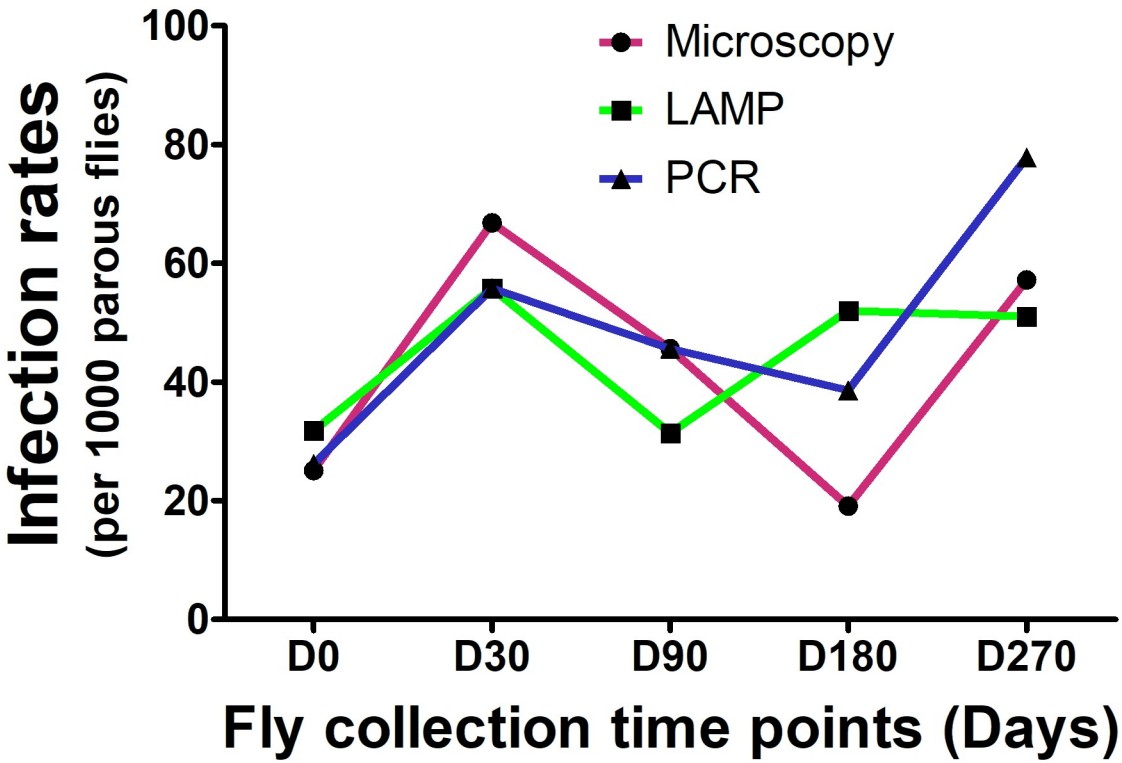

**Fig 5. Infection rates dynamics for *O. volvulus* by microscopy, LAMP and PCR.**

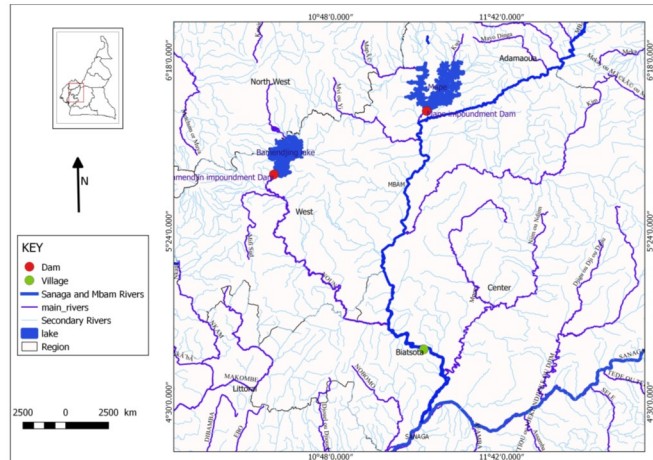

**Fig 6. Map of Mbam drainage basin showing the river system and Biatsotsa village where blackflies were collected.** This map was created using a Qgis (Qgis version 2.18.11) software.

District. The Mbam River has its origin at around 1,900m above sea level in the mountains forming the border with Nigeria to the North-North-East of Banyo in the Adamaoua Region of Cameroon. Generally, it descends rapidly towards North-South direction and after a course of 100 km, crosses West-South-West to the plain of Tikar between the coasts, 700 and 600 to the confluence of the Mape. There at Mape, it receives a series of small right bank tributaries from the border chain, including the Darie River. After the Mape, it takes the general direction North-South until the confluence of the Noun where it receives two important tributaries: the Kim River on the left bank which comes from Yoko town and the Noun River on the right bank, which is coming from the mountains close to the highlands of Mount Oku and the Ndop plains in the North West Region at an altitude of about 2,000m. The river then descends very quickly from North to South to East of Bafoussam. While the slope decreases, the river's direction bends towards the South-East, just before the town of Bafia. The Mbam River forms a very large bend radius, receives river Ndjim on its left bank and merges with Sanaga at an altitude of 380m after a course of 400 km. Its watershed at the confluence with river Sanaga is 40,000 km$^2$. The Bamendjing dam is located between longitude 10 ˚ 30 '03 " East and latitude 5 ˚ 41' 55" north, and is built on the Noun river. This reservoir dam was put into service in May 1974. It created an artificial lake whose surface varies from 25,500 ha to 33,300 ha depending on the season. The dam has a height of 22m, a surface area of 2,190 km$^2$ and the capacity of the reservoir is approximately 1.8 billion m$^3$ of water [33]. The Mape reservoir-dam, located at 11 km Northeast of Magba, on the Mape, was built in 1988 in the district of Bankim (Adamawa region, Mayo-Banyo division) and it is located at latitude 6 ˚01'56" north and longitude 11˚ 18'18" east. It has an altitude of 724m with a retention capacity of 3.2 billion m$^3$ of water on an area of 50,000 ha. The analysis of the Mbam river drainage revealed a very high rate of water discharge during both rainy and dry seasons in Bafia (study site) due to the presence of two dams on the Mbam tributaries (Noun and Mape) at Bamendjing and Mape, respectively. These retaining dams were constructed to store water during the rainy season and releases the water during the dry season to regularize the course of the Sanaga River in view of stabilizing the production of electrical energy at Edea and Son lou-lou hydro-electric dams. The high speed and volume of water plus the presence of rocks and vegetation in the river serves as excellent breeding site for blackflies in both the rainy and dry seasons of the year. Using marked positions on tree trunks and branches as reference points, we noticed very minimal

changes in water level (height) between the rainy and dry seasons within the nine months of fly collection. Fig 7 shows the view of the Mbam River at Bafia (Biatsotsa) during the dry season with an important water discharge (S1 Video) as well as the presence of larval stages of vector on vegetation in the river.

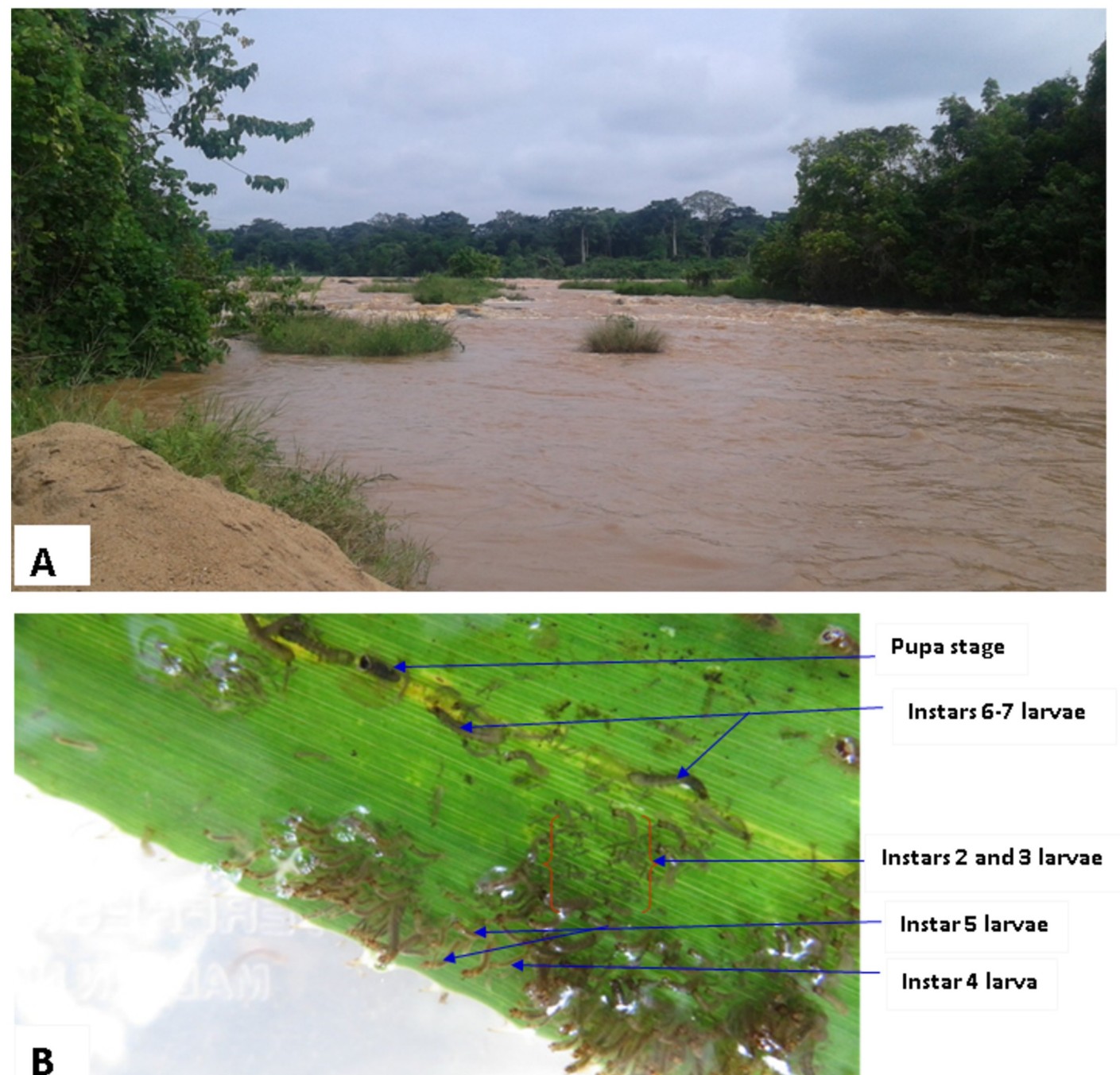

**Fig 7.  A** = View of the Mbam River at Bafia (Biatsotsa) during the dry season with an important water discharge (Source; author). **B** = Developmental stages (instars) of the *Simulium* larvae on vegetation in River Mbam at the study site; identified using Crosskey [34].

## Discussion

The presence of fast flowing rivers, abundance of blackflies with high vector capacity and efficiency, plus microfilaridemic individuals in an area will provide conditions for continuous and persistent transmission of onchocerciasis despite a long term IVM-MDA campaign.

Before and at all time points after IVM-MDA in Bafia HD, which has had more than 20 years of annual CDTI, vector abundance and high entomological indices were observed throughout the 9 months study period cutting across the rainy and dry seasons of the year. The consistently high biting rates and continuous high MTP observed throughout the study period is in agreement with the findings of Adam Hendy in the same endemic focus [6]. The abundance of vector is due to the fact that, this area is irrigated by many fast-flowing rivers (including river Mbam), which constitute breeding sites for *Simulium damnosum*. The Mbam River offers an excellent breeding site for *Simulium* in both the rainy and dry seasons of the year because of the presence of Mape and Bamendjing dams that retain water during the rainy season and releases it from their lakes during the dry season, to regularize the course of the Sanaga River in view of stabilizing the production of hydro-electrical energy at Edea. At 30 days post MDA, we observed an unexpected increase in the infection and infective rates by both microscopy and molecular techniques. This implies that, the potential reduction in Mf load following MDA in the human population was not translated to the transmission indices in the vector. This is contrary with other studies where infection and infective rates were reported to have dropped drastically within 30 to 90 days after MDA [17–19]. Given that microfilaridemia in humans infected with IVM-susceptible parasites usually drop by about 98–99% within 1 month after treatment [16], the blackflies in this study area could possess high vector capacity, capable of acquiring residual Mfs in humans and also very efficient in converting the Mfs to infective L3 larvae in order to ensure continuous transmission. From a different perspective, the parasites in this study area could have developed sub-optimal characteristics to IVM due to long exposure [35] or drug pressure as evident in the high geographical and constantly improving chemotherapeutic coverages earlier reported over the years [23,24]. In addition, the Mf load in the infected humans may not have reduced as expected, thereby keeping the possibility of vector acquiring a parasite during a blood meal unperturbed. On the other hand, selective susceptibility to IVM [36,37] may have occurred in the worm population present in this endemic focus during the first few MDA campaigns and the small proportion of non-susceptible or less susceptible Mfs was acquired by the efficient vectors and re-introduced into the human population over the years. Should this be the case, subsequent administration of IVM will not have the expected effect on the Mf load and consequently, there will always be Mfs in humans to be acquired by vectors for transmission at all seasons of the year despite the continuous CDTI campaign. Also, many of the infected humans in the study area may be among the reported less than 25% of the population that has not been complying with treatment over the years [23], thereby serving as reservoir for continuous transmission of the parasites by the abundance vector with high vector capacity and efficiency.

For the MTP, it dropped from 340 to 95 (72% reduction) within the first three months and started increasing at 6 months post MDA. This is in line with the findings of Trpis and colleagues who reported a 74.6% reduction in MTP within the first 3 months of MDA [19] The continuous transmission throughout nine months, is an indication that, IVM-MDA may have very little or no impact on the transmission profile of *O. volvulus* in the vectors found in this study area. This could be a signal of suboptimal response of the drug to the parasite when compared to previous observations where the intensity of infection in the vector populations dropped drastically within few months following IVM-MDA [17–19,38]. This continuous transmission also indicates that the present CDTI strategy for the control of onchocerciasis in

this endemic focus is not yielding expected results. The abundant vector in this area coupled with high capacity and efficiency observed in the transmission dynamics following MDA, is an indication that, even with high therapeutic coverage, blackflies in the Mbam drainage basin will make use of the least opportunity created such as presence of infected individuals who do not adhere to treatment or presence of parasites that respond sub-optimally to ivermectin either due to genetic drift [37] or due to drug pressure [35,39,40] to enhance continuous transmission.

We also observed that, all the techniques used could detect infection in the blackfly populations with very little or no difference in the sensitivity amongst the different techniques. The results are similar to that of other authors [41], in which they also reported no difference in infection rate between fly dissection and pool screen PCR. This implies that, in areas with high transmission intensity and persistent transmission, these techniques could be used interchangeably to monitor the presence of infection following MDA even though species differentiation may not be possible with microscopy. The infection and infective rates from pool screening were arrived at with the assumption of 50% parity in the vector as recommended [32]. In our study site however, we observed a lower rate of parous flies (24.7%) for the over 9000 flies dissected. If we were to consider our observed parity to compute the infection and infective rates from the pool screening, the infection and infective rates per 1000 parous flies for the molecular techniques would have doubled and automatically become much higher than those obtained by microscopy, thereby qualifying the DNA detection techniques (real-time PCR and LAMP) to be comparable but more sensitive than microscopy.

Though at a very low rate, *O. ochengi* parasites were detected in this study area in the month of April 2017 (9 months after IVM-MDA), implying that, the *Simulium* vectors in this study areas are capable of acquiring other *Onchocerca species* and they may or may not develop to infective larvae (L3H). Important to note that we had primers to investigate *O. ochengi* but did not have primers for other animal species such as *O. ramchandani*. Hence microscopy should always be complemented by real-time PCR or LAMP, in order to detect other *Onchocerca species in flies*.

After more than 20 years of implementing CDTI in Bafia HD, the expected results (infective rate of less than or equal to 0.1% in 1000 parous flies or 0.05% in all flies, assuming a parity of 50%) required to stop IVM-MDA have not been attained [42]. There is need to employ alternative strategies endorsed by the World Health Organization such as test-and-treat with doxycycline macrofilaricide which targets and kills the adult parasites in humans [43,44] and ground larviciding with organophosphate temefos for suppression of blackfly vectors [45], in order to accelerate onchocerciasis elimination in the Mbam drainage basin and the extended transmission zones including the Massangam Health District of West Region around River Nja (a tributary of River Noun) and River Kichi (a tributary of River Mbam) [46]. These alternative strategies are already being implemented to accelerate onchocerciasis elimination in an area of loiasis co-endemicity in the South West Region of Cameroon [47].

## Conclusion

In this area of persistent transmission of onchocerciasis of the Mbam river basin, we observed that *Simulium* flies are highly abundant throughout the year. Blacklies could still acquire microfilariae from the infected human host despite the IVM-MDA to the population. MDA did not prevent the intake of *O. volvulus* Mf from the human host and did not interrupt the development of Mf into infective larvae in the vectors. Microscopy, real-time PCR and LAMP revealed the same trends in the detection of *O. volvulus* infection levels in *Simulium* population pre and post MDA. The infection rate with animal *Onchocerca sp* was exceptionally low,

confirming that the larvae observed by microscopy in this area could be mostly *O. volvulus*. We also observed excellent breeding sites for *Simulium* during both dry and rainy seasons, facilitated by the release of water from the two retention dams during the dry season. Despite the reported high geographical (99.5%) and improved therapeutic (from 42% to over 80%) coverages of MDA in this area over the years, the high vector abundance (enhanced by the river drainage system) with high vector capacity and efficiency have been able to continuously transmit the parasite from the few infected individuals among those who do not comply to treatment (or the few individuals carrying parasite strains that respond sub-optimally to ivermectin) to non-infected individuals, to maintain the observed uninterrupted transmission profile before and after MDA campaign. There is need for alternative control strategies such as ground larviciding with temephos and test-and-treat with doxycycline to accelerate the fight against onchocerciasis in this drainage basin.

## Supporting information

**S1 File. Dissection sheets.**
(XLS)

**S2 File. Real-time PCR and LAMP results for pool screening.**
(XLS)

**S1 Fig. Check for inhibition factors.**
(TIF)

**S2 Fig. Representative data obtained from pool samples using a colorimetric LAMP.** A.). Samples containing *O. volvulus* larval stage(s) turned yellow and were scored positive (i.e. tubes -2-, -3-, -7- and -8-). Negative pools remained pink (i.e. tubes -1-, -4-, -5- and -6-). B.). Specificity of colorimetric LAMP assay confirmed. Reaction contained no template DNA (> or tube -9-) or DNA from *Onchocerca volvulus* (OV or tube -10-), or *Mansonella perstans* (MP or tube -11-) or *Loa loa* (LL or tube -12-).
(TIF)

**S1 Video. Short video of water discharge at Biatsotsa during the dry season.**
(MP4)

## Acknowledgments

We are grateful to the fly collectors from the village of Biatsota in the Bafia health district who willingly accepted to participate in this study. Many thanks to the Chief of Biatsota village, chief of health center, District Medical officer and Chief of Bureau health, the Regional Delegate of Public Health for the Center Region, and the Regional and National Coordinators of onchocerciasis control programme, for their administrative assistance. We are most grateful to New England Biolabs (NEB) in USA and Institute for Medical Microbiology, Immunology and Parasitology, University Hospital Bonn, Germany, for providing all the reagents for this work.

## Author Contributions

**Conceptualization:** Raphael Awah Abong, Catherine Poole, Kenneth Pfarr, Achim Hoerauf, Clotilde Carlow, Samuel Wanji.

**Data curation:** Raphael Awah Abong, Glory Ngongeh Amambo, Ali Ahamat Hamid, Belinda Agbor Enow, Amuam Andrew Beng, Theobald Mue Nji, Abdel Jelil Njouendou, Fanny Fri Fombad, Peter Ivo Enyong, Samuel Wanji.

**Formal analysis:** Raphael Awah Abong, Franck Noel Nietcho, Abdel Jelil Njouendou, Samuel Wanji.

**Investigation:** Raphael Awah Abong, Glory Ngongeh Amambo, Ali Ahamat Hamid, Belinda Agbor Enow, Amuam Andrew Beng, Theobald Mue Nji, Abdel Jelil Njouendou, Manuel Ritter, Fanny Fri Fombad, Peter Ivo Enyong, Samuel Wanji.

**Methodology:** Raphael Awah Abong, Glory Ngongeh Amambo, Catherine Poole, Kenneth Pfarr, Samuel Wanji.

**Resources:** Samuel Wanji.

**Supervision:** Catherine Poole, Kenneth Pfarr, Samuel Wanji.

**Validation:** Raphael Awah Abong, Kenneth Pfarr.

**Writing – original draft:** Raphael Awah Abong, Samuel Wanji.

**Writing – review & editing:** Raphael Awah Abong, Glory Ngongeh Amambo, Ali Ahamat Hamid, Belinda Agbor Enow, Amuam Andrew Beng, Theobald Mue Nji, Abdel Jelil Njouendou, Manuel Ritter, Mathias Eyong Esum, Kebede Deribe, Jerome Fru Cho, Fanny Fri Fombad, Peter Ivo Enyong, Catherine Poole, Kenneth Pfarr, Achim Hoerauf, Clotilde Carlow, Samuel Wanji.

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
