## [Decision Letter · Decision Letter 0]

28 Aug 2020

Dear Prof. Wanji,

Thank you very much for submitting your manuscript "Observations on the hydrologic network and dynamics of onchocerciasis transmission within nine months following a mass drug administration campaign with ivermectin in the Mbam river drainage, a site with persistent transmission in Cameroon" for consideration at PLOS Neglected Tropical Diseases. As with all papers reviewed by the journal, your manuscript was reviewed by members of the editorial board and by several independent reviewers. The reviewers appreciated the attention to an important topic. Based on the reviews, we are likely to accept this manuscript for publication, providing that you modify the manuscript according to the review recommendations. 

Sincerely,

Wilma A. Stolk, Ph.D.

Guest Editor

Jaap van Hellemond

Deputy Editor

Reviewer's Responses to Questions

**Key Review Criteria Required for Acceptance?**

**Methods**

-Are the objectives of the study clearly articulated with a clear testable hypothesis stated?

-Is the study design appropriate to address the stated objectives?

-Is the population clearly described and appropriate for the hypothesis being tested?

-Is the sample size sufficient to ensure adequate power to address the hypothesis being tested?

-Were correct statistical analysis used to support conclusions?

-Are there concerns about ethical or regulatory requirements being met?

Reviewer #1: The methods were presented

Reviewer #2: 1. The objectives are clearly articulated

2. The study design is consistent with expected standard

3.The population is clearly described and appropriate

4. The sample size is sufficient

5. Statistical analysis is correct to support the conclusion 

6. There are no concerns about ethical or regulatory requirement

Reviewer #3: The methods are articulated well and the hypothesis is clearly stated

The study design is appropriate 

The population is clearly decribed

The sample size is sufficient

The statistics is fine 

There are no concerns about ethical or regulatory requirements

**Results**

-Does the analysis presented match the analysis plan?

-Are the results clearly and completely presented?

-Are the figures (Tables, Images) of sufficient quality for clarity?

Reviewer #1: The analysis is fine

Reviewer #2: Presentation of the result is satisfactory and consistent with expected standard. However, the results of the revised manuscript should factor in the geographical and chemotherapeutic coverage of the MDA in the past 15 years of the transmission zone

Reviewer #3: The analyses presents matches the plan

The results are clearly presented

Tables and Figures are sufficient

**Conclusions**

-Are the conclusions supported by the data presented?

-Are the limitations of analysis clearly described?

-Do the authors discuss how these data can be helpful to advance our understanding of the topic under study?

-Is public health relevance addressed?

Reviewer #1: The conclusion if fine

Reviewer #2: 1There is need for the introduction of the geographical and chemotherapeutic coverage of the MDA in the past 15 years of the transmission zone and its impact on the dynamics of Onchocerca volvulus infection in the black flies in the conclusion. 

2. The authors discussed how these data can be helpful to advance our understanding of the topic under study

3. The public health relevance of the study was addresses

Reviewer #3: The conclusion support the data presented

The limitations are fine

The author does describe how this data can advance entomology

The public health significance is discussed

**Editorial and Data Presentation Modifications?**

Reviewer #1: Minor Revision

Reviewer #2: 1. Title: The title is good, relevant and operational but I suggest that it can be made smarter by reframing as “Hydrologic network and onchocerciasis transmission post ivermectin MDA campaign in the Mbam river drainage, Cameroon”

2. Abstract; Consistent with expected standard but the following corrections on RED can be effected to add advantage to it:

Introduction: The impact of large scale Mass Drug Adminstration (MDA) of ivermectin on Simulium damnosum, which transmits the parasite Onchocerca volvulus is of great importance for onchocerciasis control programmes. We investigated the impact of MDA of ivermectin on entomological indices and also verify if there are hydrologic network factors that could have favoured the transmission of onchocerciasis in this area and contribute to the persistence of disease. We compared three independent techniques to detect Onchocerca larvae in blackflies and also analyzed the hydrologic network within 9 months post-MDA of ivermectin.

Method: Methodology is consistent with expected standard and the experimental design is satisfactory.

Results: Satisfactory but the underlisted correction should be effected:

Line 49: We used human landing collection method (previously called human bait) to collect 22,274 adult female Simulium flies from Mba River System. Of this number, 9,134 were dissected while 129 pools constituted for

Conclusion: Satisfactory but the under listed corrections should be effected:

Line 56: Results from fly dissection (Microscopy), real-time PCR and LAMP revealed the same trends 

Key words: Mbam river drainage, MDA Ivermectin Simulium, biting rate, infection rate

2. Introduction: The under listed corrections should be effected:

Line 82-83: Onchocerciasis is a parasitic disease caused by Onchocerca volvulus, transmitted by the bite of an infective female

Line 92: The control/elimination of onchocerciasis today relies on ivermectin (Mectizan®) mass drug administration

Line 95: programmes. Mectizan® has been donated free of charge by Merck and Co.

Line 96/97: Whitehouse Station, NJ since 1987. Ivermectin must be repeatedly given once a year for more than 15 years (with at least 60% geographical and chemotherapeutic coverage of MDA) to successfully achieve the elimination of onchocerciasis by the year 2030

Line 128/129: Following the adoption of IVM-MDA as a control/eliminaltion strategy in Cameroon, it has been demonstrated that the Community Directed Treatment with Ivermectin (CDTI) project of Central Region at the level of Bafia had limited impact on 

Line 134: Delete (or Mf are not being cleared in the treated individuals) 

Line 137: vector capacity and efficiency of Simulium damnusum for the transmission of onchocerciasis in this area and 

Lines 139-145: Should be reframed to reflect the objectives of the study not the procedure of study

3. STUDY AREA: Effect the following correction:

Lines 170-177: The geographical and chemotherapeutic coverage of the MDA in the past 15 years of the transmission zone should be included

Line 181: Collection of Simulium flies by human landing method

Lines 207 – 272 covering the following should be reframed and summarized.

(i) Genomic DNA extraction from pools of Simulium flies, 

(ii) (ii) Detection of O. volvulus in pool-DNA of blackfly using colorimetric O-150 LAMP and 

(iii) Detection of O. volvulus / O. ochengi in pool-DNA of blackfly using real-time PCR 

4. RESULT: Presentation of the result is satisfactory and consistent with expected standard. However, the results of the revised manuscript should factor in the geographical and chemotherapeutic coverage of the MDA in the past 15 years of the transmission zone

5. DISCUSSION: The result of this study is well discussed but the absence of the data on the impact of geographical and chemotherapeutic coverage of the MDA in the past 15 years on the transmission zone should be taken into consideration.

6. CONCLUSION: There is need for the introduction of the geographical and chemotherapeutic coverage of the MDA in the past 15 years of the transmission zone and its impact on the dynamics of Onchocerca volvulus infection in the black flies in the conclusion.

7. REFERENCES: The authors demonstrated an in-depth knowledge of the field of study and also made due acknowledgement of existing information. However, there are minor collections that should be effected:

(a) The biological names in reference numbers 20, 25, 30, 35, 36 and 38 should be in Italics

(b) Reference 32 should re-written to be consistent with the expected format.

8. OBSERVATIONS

(a) This study has made original contributions to scientific knowledge by bringing to focus the impact of hydrologic network/characteristics of transmission zone and other factors on onchocerciasis transmission post-mass MDA of ivermectin 

(b) The introduction of the geographical and chemotherapeutic coverage of the MDA in the past 15 years of the transmission zone and its impact on the dynamics of Onchocerca volvulus infection in the black flies is going to add value to the work.

9. RECOMMENDATIONS

(a) The manuscript should be accepted for publication but NOT in its present form

(b) The corrections pointed out should be effected before publications, especially the introduction of the treatment coverage (both geographical and chemotherapeutic coverage) of the MDA in the past 15 years of the transmission zone and its impact on the dynamics of Onchocerca volvulus infection

Thanks for inviting me to serve and contribute.

Please accept the continued assurances of my highest esteem and regards.

Prof. B. E. B. Nwoke

Reviewer #3: Here are my views….

36..Impact of large scale ivermectin MDA on s. damnosum……. This is as though Ivermectin had direct impact on s. damnosum…In that sense, there would be need to have entomological proof. Instead, it would be….Impact of large scale ivermectin MDA on active onchocerciasis transmission. Your investigations were based on the presence of O. Volvulus lavae in the fly, and rather not the physical health of the fly.

65.. The presence of parasite strains that respond sub-optimally to an approved drug….With regular ivermectin MDA, its known that host parasite load is significantly reduced and consistent ivermectin MDA can cut oncho transmission even amidst abundance of s. damnosum biting….So,. ivermectin is not sub-optimally effective against microfilaria……..To dispute ivermectin, investigate and show proof of community compliance to ivermectin uptake.

69... The hydrologic network factors that could have favored abundance vector breeding and contributed to the persistence of disease transmission were also examined within the study period….I think what you exactly did here was a study of the mbam river drainage system. 

92.. The control of onchocerciasis today relies on ivermectin mass drug administration (IVM-MDA) to at risk populations in endemic countries….If this is true then ivermectin is not sub-optimally effective against micro filaria

**Summary and General Comments**

Reviewer #1: This is a well thought out study

Reviewer #2: (a) The manuscript should be accepted for publication but NOT in its present form

(b) The corrections pointed out should be effected before publications, especially the introduction of the treatment coverage (both geographical and chemotherapeutic coverage) of the MDA in the past 15 years of the transmission zone and its impact on the dynamics of Onchocerca volvulus infection

Reviewer #3: The paper is fine. Its significance is important in terms of implementing larval control for this vector

PLOS authors have the option to publish the peer review history of their article (what does this mean?). If published, this will include your full peer review and any attached files.

Reviewer #1: Yes: Moses Katabarwa

Reviewer #2: Yes: Prof. Bertram Ekejiuba Bright NWOKE

Reviewer #3: No
---

## [Editor Report · Decision Letter 1]

17 Oct 2020

Dear Prof. Wanji,

Thank you very much for submitting your manuscript "The Mbam drainage system and onchocerciasis transmission post ivermectin Mass Drug Administration (MDA) campaign, Cameroon" for consideration at PLOS Neglected Tropical Diseases. As with all papers reviewed by the journal, your manuscript was reviewed by members of the editorial board and by several independent reviewers. Thank you for addressing their comments. The reviewers appreciated the attention to an important topic. Based on the reviews, we are likely to accept this manuscript for publication, providing that you modify the manuscript according to the editorial recommendations listed below. 

Sincerely,

Wilma A. Stolk, Ph.D.

Associate Editor

Jaap van Hellemond

Deputy Editor

Editorial suggestions (line numbers refer to the clean version of revision 1)

1) References 3 and 5 are no longer available online. Please replace by more up to date references.

2) Abstract, line 37: make clear that this work is done in the Mbam river system area

3) Methods, study area, lines 164-166: clarify how coverage was calculated in this case. If I am correct, you are referring to reported coverage levels, expressed as number treated out of the total population. Ref 24 also highlights that survey coverage estimates are considerably lower than the reported ones. This should be acknowledged. Check whether similar changes are required in the concluding paragraph of the paper.
---

## [Editor Report · Decision Letter 2]

27 Oct 2020

Dear Prof. Wanji,

We are pleased to inform you that your manuscript 'The Mbam drainage system and onchocerciasis transmission post ivermectin Mass Drug Administration (MDA) campaign, Cameroon' has been provisionally accepted for publication in PLOS Neglected Tropical Diseases.

Best regards,

Wilma A. Stolk, Ph.D.

Associate Editor

Jaap van Hellemond

Deputy Editor

---

## [Editor Report · Acceptance letter]

31 Dec 2020

Dear Prof. Wanji,

We are delighted to inform you that your manuscript, "The Mbam drainage system and onchocerciasis transmission post ivermectin Mass Drug Administration (MDA) campaign, Cameroon," has been formally accepted for publication in PLOS Neglected Tropical Diseases.

Best regards,

Shaden Kamhawi

co-Editor-in-Chief

Paul Brindley

co-Editor-in-Chief
